# OceanSODA-UNEXE: A multi-year gridded Amazon and Congo River outflow surface ocean carbonate system dataset

Richard P. Sims[1], Thomas M. Holding[2], Peter E. Land[3], Jean-Francois Piolle[4], Hannah L. Green[1,3], Jamie D. Shutler[1]

[1]Centre for Geography and Environmental Science, College of Life and Environmental Sciences, University of Exeter, Penryn campus, United Kingdom

[2]Department of Human Behaviour, Ecology and Culture, Max Planck Institute for Evolutionary Anthropology, Leipzig, 04103, Germany

[3]Plymouth Marine Laboratory, Plymouth, PL13DH, United Kingdom

[4]Laboratoire d'Océanographie Physique et Spatiale (LOPS), IFREMER, Université of Brest, CNRS, IRD, IUEM, Brest, France

*Correspondence to*: Richard P. Sims (r.sims2@exeter.ac.uk)

## Abstract

Large rivers play an important role in transferring water and all of its constituents including carbon in its various forms from the land to the ocean, but the seasonal and inter-annual variations in these riverine flows remain unclear. Satellite Earth observation datasets and reanalysis products can now be used to observe synoptic-scale spatial and temporal variations in the carbonate system within large river outflows. Here we present the OceanSODA-UNEXE time series, a dataset of the full carbonate system in the surface water outflows of the Amazon (2010-2020) and Congo Rivers (2002-2016). Optimal empirical approaches were used to generate gridded Total alkalinity (TA) and dissolved inorganic carbon (DIC) fields in the outflow regions. These combinations were determined by equitably evaluating all combinations of algorithms and inputs against a reference matchup database of *in situ* observations. Gridded TA and DIC along with gridded temperature and salinity data enable the calculation of the full carbonate system in the surface ocean (which includes pH and the partial pressure of carbon dioxide, $p$CO$_2$). The algorithm evaluation constitutes a Type A uncertainty evaluation for TA and DIC where model, input and sampling uncertainties are considered. Total combined uncertainties for TA and DIC were propagated through the carbonate system calculation allowing all variables to be provided with an associated uncertainty estimate. In the Amazon outflow, the total combined uncertainty for TA was 36 μmol kg$^{-1}$ (weighted RMSD 35 μmol kg$^{-1}$ and weighted bias 8 μmol kg$^{-1}$ for n=82) and for DIC was 44 μmol kg$^{-1}$ (weighted RMSD 44 μmol kg$^{-1}$ and weighted bias -6 μmol kg$^{-1}$ for n=70). The spatially averaged propagated combined uncertainties for the $p$CO$_2$ and pH are 85 μatm and 0.08 respectively, where the pH uncertainty is relative to an average pH of 8.19. In the Congo outflow, the combined uncertainty for TA was identified as 29 μmol kg$^{-1}$ (weighted RMSD 28 μmol kg$^{-1}$and weighted bias 6 μmol kg$^{-1}$ for n=102) and for DIC was 40 μmol kg$^{-1}$ (weighted RMSD 37 μmol kg$^{-1}$and weighted bias -16 μmol kg$^{-1}$ for n=77). The spatially averaged propagated combined uncertainties for $p$CO$_2$ and pH are 74 μatm and 0.08 respectively, where the pH uncertainty is relative

to an average pH of 8.21. The combined uncertainties in TA and DIC in the Amazon and Congo outflows are lower than the natural variability within their respective regions allowing the time varying regional variability to be evaluated. Potential uses of these data would be for assessing the spatial and temporal flow of carbon from the Amazon and Congo rivers into the Atlantic and for assessing the riverine driven carbonate system variations experienced by tropical reefs within the outflow regions.

**1 Introduction**

Rivers connect the land and ocean providing a major pathway for carbon transport to the ocean (Regnier et al., 2013). The inorganic carbon content of rivers is poorly constrained because it is difficult to sufficiently sample these highly spatial and temporally variable river outflows. Global estimates of the riverine flow of carbon from the land to the ocean (Friedlingstein et al., 2022) are determined from *in situ* upscaling (Regnier et al., 2013), ocean inverse model estimates (Jacobson et al., 2007), partial pressure of carbon dioxide ($pCO_2$) ocean sink based estimates (Watson et al., 2020) and atmosphere inversion based estimates (Rödenbeck et al., 2018). Clearly new methods that improve the characterisation of the magnitude, variability and temporal variations of carbon transported by rivers will help constrain uncertainties within global carbon budgets (Hauck et al., 2020).

The vast majority of $pCO_2$ measurements in the Surface ocean $CO_2$ Atlas (SOCAT) were made on research ships and ships of opportunity that mainly survey the open ocean and continental shelves (Bakker, 2016). Many bottle samples have been collected for total alkalinity (TA) and dissolved organic carbon (DIC) in rivers, but fully surveying the entirety of large rivers across all seasons is logistically challenging requiring extensive and economically expensive field campaigns (Ward et al., 2017). For example, the majority of the largest 100 rivers by discharge are found in South America and Asia (Dai and Trenberth, 2002) and have been historically under-sampled for carbonate system variables (Laruelle et al., 2015). Issues related to scarcity of measurements is compounded by insufficient knowledge of the hydrology and spatial area extents of these systems (Allen and Pavelsky, 2018). Additionally, the amount of carbon in the rivers is a function of runoff rates, rainfall and land use, all of which have been disrupted by climate change and land use change (Piao et al., 2007;Kaushal et al., 2014;Regnier et al., 2013). This lack of large spatial and temporal scale baseline carbonate system observations in rivers means that assessing changes in these systems is challenging.

The carbon dynamics of the world's rivers also have implications on local biogeochemistry. Ocean acidification is the long-term process by which the oceans absorb atmospheric $CO_2$ making them less alkaline due to an increase in the hydrogen ion concentration, lowering their pH and decreasing the carbonate ion availability (Doney et al., 2009). Ocean acidification poses a threat to marine organisms which build calcium carbonate structures and many rivers are sensitive to this due to their

low buffering capacity (Hu and Cai, 2013;Cai et al., 2011). River plumes can negatively influence wild fisheries and the aquaculture industry (Mathis et al., 2015;Cattano et al., 2018) as plumes can transport low pH waters with the ability to impact the growth and life stages of many marine organisms (Cai et al., 2021). Additionally, river plumes can interact with high biodiversity regions that are sensitive to sudden changes in the carbonate system such as sensitive coral reef systems (Mongin et al., 2016;Dong et al., 2017). Intermittent changes in the carbonate system caused by river plumes can potentially jeopardise ecosystem services like fisheries, aquaculture, and shoreline protection, the resultant financial and biodiversity losses and of great interest to local communities, businesses and policy makers (Doney et al., 2020).

Satellite Earth observations provide a means of accurately assessing the carbonate content of large rivers by using satellite observed oceanographic variables with published empirical algorithms that link carbonate system variables to the satellite derived variables (Land et al., 2019). The Satellite Oceanographic Datasets for Acidification (OceanSODA) project (https://esa-oceansoda.org) was established to further develop these approaches. We present the University Of Exeter (UNEXE) OceanSODA dataset (OceanSODA- UNEXE), decadal datasets of riverine carbonate system variables for the two largest rivers in the world by discharge, the Amazon and the Congo Rivers (Dai and Trenberth, 2002). This paper details how the optimal combination of empirical algorithms and Earth observation datasets were selected and used to construct OceanSODA-UNEXE. This paper provides an assessment of the uncertainty associated with the key TA and DIC parameters using a standardised uncertainty framework and a large *in situ* database. The remaining ocean carbon system variables were calculated from TA and DIC with propagated uncertainties.

## 2 Methods

### 2.1 Statistical terms overview

Root mean squared difference (RMSD) is a measure of accuracy and is calculated as the square root of the average of squared errors e.g. RMSD = $((\Sigma(x_0 - x_1)^2)/n)^{1/2}$, where $x_0$ are the estimated values, $x_1$ are the reference values and n is the number of observations. The bias of a dataset is defined as the mean difference between the estimation and reference, Bias= $\Sigma(x_0 - x_1))/n$. The mean absolute difference (MAD) of a dataset is a measure of variability and is calculated as MAD = $(\Sigma|x - \bar{x}|)/n$, where $\bar{x}$ is the mean. The correlation coefficient ($r_{ij}$) is a measure of linear correlation between the estimate and the reference variables and is defined as $r_{ij} = (\Sigma (x_i - \bar{x}_i) (x_j - \bar{x}_j)) / (\Sigma (x_i - \bar{x}_i)^2 \Sigma (x_j - \bar{x}_j)^2)^{1/2}$. Uncertainty representation and the terminology used throughout this paper are consistent with the International Bureau of Weights and Measures (BIPM) Guide to the expression of uncertainty in measurement (GUM) methodology (JCGM, 2008).

Weighted statistics allow uncertainties in the reference dataset (*in situ* measurement uncertainty in this case) to be accounted for within the performance analysis (i.e. the reference data are not considered 'truth' as they also contain uncertainties). Weights are calculated as the sum of the individual weight of each algorithm (w), where w = $1/((in\ situ$ measurement uncertainty)$^2$+(literature algorithm uncertainty)$^2$)$^{1/2}$ (Ford et al., 2021). For clarity and easy comparison to previous published work both weighted and unweighted statistics of all metrics are given, for example weighted RMSD (wRMSD) is calculated as wRMSD= $(\Sigma(weights*(x_0 - x_1)^2))^{1/2}$.

When evaluating algorithms using a statistical measure, in this case wRMSD, a further issue can arise where the valid region over which each algorithm can be applied overlaps with different *in situ* data. For example, an algorithm evaluated using data from highly-variable coastal waters may have a higher wRMSD than another algorithm evaluated using solely data from a less variable open ocean region; in this scenario it may be falsely concluded that the coastal ocean algorithm performs worse. This is a clear weakness of comparing wRMSD values from different sources and across differing regions (Land et al., 2019). Following the methodology of Land, Findlay et al. (2019) we calculate RMSDe using the wRMSD result, RMSDe is a more representative metric to compare accuracies as it allows algorithms to be evaluated in a like for like manner allowing their performance to be ranked. It is important to note that for the best algorithm, wRMSD = RMSDe as the best algorithm will always have a score of 1. To ensure the robustness of any statistics generated it was considered prudent to specify a minimum data threshold. Only algorithms that had at least 30 matchups (n=30) between the algorithm and reference outputs were used in the calculation of RMSDe, this was done to prevent the selection of algorithms with low RMSDe values caused by evaluating the algorithm with a small number of data points. wRMSD is used as the preferred measure of accuracy in this paper but unweighted RMSD values are also given.

## 2.2 Selection of empirical algorithms

In order to generate the full ocean carbonate system, two of the four carbonate system variables are needed. As TA is closely linked to salinity it is selected as one variable, for the second variable DIC is selected. There are many algorithms in the published literature for both TA and DIC and the required measurements in the river outflows are available to evaluate those algorithms. Other pairings of carbonate system parameters can be used to derive the full carbonate system (Land et al., 2015) but are not explored here, for example . Gregor and Gruber (2021) use TA and $pCO_2$.

An exhaustive literature search using 24 search terms identified prospective algorithms that could be applied to the Amazon and Congo River outflows. The full list of search terms and identified algorithms can be found in the supplementary materials. The region bounds of the Amazon outflow were defined as being 2° S to 24° N and 70° W to 31° W. The bounds of the Congo outflow were defined as being between 10°S to 4°N and 2° W to 16° E. To be included in the algorithm evaluation, algorithms needed to be applicable to these regions and to take the form of a linear or quadratic relationship with

input variables that were easy to obtain and available as spatially and temporally varying datasets. These input variables included sea surface temperature (SST), sea surface salinity (SSS), potential temperature (which is assumed to be approximately equal to SST at the surface), dissolved oxygen (DO), nitrate ($NO_3^-$), phosphate ($PO_4^{-3}$), silicate ($SiO_4^{-4}$) and chlorophyll-a. 26 of the identified algorithms were not included in the algorithm evaluation as they could not generate TA or DIC using the accessible input variables listed above or were not based on empirical algorithms (See Table S2 for the full list). Any approaches involving biogeochemical models or neural networks were not included in the algorithm evaluation. 10 TA algorithms (5 of 10 report RMSD values in their original publication) and 6 DIC algorithms (5 of 6 report RMSD) were evaluated for the Amazon and 4 TA algorithms (3 of 4 report RMSD) and 9 DIC algorithms (4 of 7 report RMSD) were evaluated for the Congo (Table S1). wRMSD and RMSDe can only be calculated for algorithms which report RMSD, so it is important to distinguish these. Table S1 details each algorithm's input variables, the published algorithm RMSD, the stated environmental ranges for which the algorithm is valid and some brief descriptive notes of how the algorithm was developed. The target output variable, the input variables, the mathematical algorithm, the valid geographical region and the valid geophysical conditions for each algorithm were then gathered for use in the algorithm evaluation process.

## 2.3 Algorithm evaluation

A multipurpose global reference matchups database (MDB) matching *in situ* carbonate system parameters from the surface 10 m with satellite, model and interpolated *in situ* datasets was used to perform the algorithm evaluation (Land et al., 2023). The MDB is optimised to reduce biases arising from uneven data density; this is achieved by grouping *in situ* observations into 100 km diameter regions of interest (ROI) which span a 10 day time period. The MDB was constructed from global datasets that include all of the variables required for all algorithms (SST, SSS, θ, DO, $NO_3^-$, $PO_4^{-3}$, $SiO_4^{-4}$ and chlorophyll-a see section 2.2). The MDB includes three global SST datasets: European Space Agency Climate Change Initiative SST (ESACCI SST) v2.1 (Merchant et al., 2019;Good et al., 2019), Optimum Interpolation SST (OISST) v2.1 (Huang et al., 2021;Banzon et al., 2016) and the Coriolis Ocean dataset for Reanalysis (CORA) v5.2 (Szekely et al., 2019)) and four global SSS datasets: European Space Agency Climate Change Initiative SSS (ESACCI SSS) v2.31 (Boutin et al., 2020;Boutin et al., 2021), CORA v5.2 (Szekely et al., 2019), the *in situ* Analysis System (ISAS 15 ) (Kolodziejczyk et al., 2021;Gaillard et al., 2016) and Remote Sensing Systems data from the Soil Moisture Active Passive satellite (RSS-SMAP) level 3 v4.0 (Meissner et al., 2019;Meissner et al., 2018)). The MDB also contains TA, DIC and pH values, most of which come from the Global Ocean Data Analysis Project (GLODAPv2.2020) database (Olsen et al., 2016) and $pCO_2$ values most of which come from SOCAT v2020 (Bakker, 2016) along with additional data for Arctic waters. A full list of references for data variable sources used in the MDB can be found in (Land et al., 2023). Before beginning this analysis we limited our analysis to only used MDB data that fall within these bounds: SST>-10 °C or <40 °C, SSS>0 or <50, DIC>500 μmol kg$^{-1}$ or <3000 μmol kg$^{-1}$, TA>500 μmol kg$^{-1}$ or <3000 μmol kg$^{-1}$, pH >6 or <8.5 and $pCO_2$>100 μatm or <3000 μatm and these constraints are consistent with the conditions likely within the Amazon and Congo regions.

Each valid algorithm was implemented by closely following the specific literature recommendations; this was done in such a way that empirical algorithms were only assessed for the geographical and geophysical ranges for which they were originally developed. The algorithm evaluation involved running each algorithm with inputs from the MDB to estimate the TA and DIC outputs (termed the "algorithm output"). Each algorithm output was then evaluated using the TA and DIC values from the MDB matchup database (termed the "reference output"). The RMSD, bias, correlation coefficient, and MAD are calculated from the algorithm output and the reference output. Additionally, wRMSD was calculated for algorithms where an RMSD is reported in the literature. Where wRMSD was calculated and the matchups (n) were >30, RMSDe was calculated for those algorithms following the methodology of Land et al. (2019). As the MDB includes three global SST and four global SSS datasets, each algorithm was run for each of the twelve combinations of SST and SSS input datasets, and separate statistics were generated for each configuration. The subset of input variables from the MDB used to implement the algorithm, the algorithm output and the reference output for all 12 SST and SSS combinations are included in the supplementary materials. Additionally, summary statistics between the algorithm output and the reference output are also provided in the supplementary materials, these statistics are the mean, wmean, standard deviation, RMSD, wRMSD, RMSDe, Bias, wBias, r, wr, MAD and wMAD.

The best performing TA and DIC algorithms for both regions were determined by ranking the algorithms by RMSDe, the best performing algorithms were the algorithms with the lowest RMSDe values. In order to select algorithms suitable for generating time series, observation datasets for SSS and SST needed a temporal overlap of at least 8 years. The remainder of this paper exclusively discusses the time series generated with these "optimal algorithms".

The algorithm evaluation process follows a standardised framework for a Type A uncertainty evaluation for TA and DIC whereby the following sources of uncertainties are considered:

1. The TA or DIC measurement uncertainty. The measurement uncertainty for both TA and DIC is typically stated as $\pm 4$ $\mu$mol kg$^{-1}$ but following (Bockmon and Dickson, 2015), we use their more conservative measurement uncertainty of $\pm 10$ $\mu$mol kg$^{-1}$ (approximately equal to 0.5% of the nominal TA and DIC values for the open ocean);
2. The algorithm uncertainty which is the RMSD stated in the literature for each algorithm;
3. The spatial uncertainty arising from spatial heterogeneity in the region;
4. The uncertainty due to differences in the measurement depths in the MDB.

The measurement uncertainty of 10 $\mu$mol kg$^{-1}$ and algorithm uncertainty are explicitly considered in the calculated of wRMSD and the remaining uncertainties should be minimised through using the MDB which was designed to minimise these components (Land et al., 2023). By accounting for all known sources of uncertainty this constitutes a complete Type A uncertainty evaluation. The combined standard uncertainty ($\delta Q$) for TA and DIC is calculated as $\delta Q = ((\delta a)^2 + (\delta b)^2)^{1/2}$ where

δa is the RMSDe from the evaluation algorithm and δb is the wBias. The uncertainty estimates provided within the dataset is the combined standard uncertainty (δQ) and these combined standard uncertainties are reported without confidence intervals.

To aid interpretation of the combined uncertainty budgets, a second Type A uncertainty evaluation of the TA and DIC approaches was calculated based purely on the empirical algorithm (e.g. literature RMSD value) and the uncertainties in the input data propagated using standard techniques (Taylor, 1997) and assuming that any uncertainties were uncorrelated. This is a second independent Type A uncertainty evaluation that does not include uncertainties due to spatial variability and depth. Assuming that the first Type A uncertainty evaluation has captured all the uncertainties, the difference between the two uncertainty evaluations enables estimation of the contribution of the spatial and depth variability on the uncertainty budget. A similar approach was recently performed by Gregor and Gruber (2021) where they refer to these two approaches as 'top down' (our first Type A uncertainty evaluation) and 'bottom up' (our second Type A uncertainty evaluation) assessments.

Evaluating each algorithm exactly as it appears in the literature is required to ensure equitably in the algorithm evaluation. Theoretically, all of the algorithms could have been modified to account for long-term changes in environmental conditions (e.g. oceanic $CO_2$ uptake and increased freshwater input to the oceans) in these two outflow regions that have likely occurred since the algorithms were first developed. However, accounting for changing environmental conditions is not straightforward. Whilst secular trends for variables like surface $CO_2$ are accurate for the open ocean, these regions are heavily impacted by the river outflows and so applying open ocean secular trends to these riverine regions is likely problematic and so considered beyond the scope of this study. The impact of any secular trends in any parameters is somewhat mitigated as the analysis focuses on the use of TA and DIC as these variables are likely less impacted by secular trends than $p$CO$_2$ and pH.

## 2.4 Creation of gridded monthly time series product

The full spatiotemporal resolution SST and SSS datasets and gridded World Ocean Atlas (WOA) DO, $NO_3^-$, $PO_4^{-3}$ and $SiO_4^{-4}$ datasets were downloaded from their online repositories, the total size of these full resolution datasets occupies ~500 GB of hard drive space and take several days to fully download. The SST and SSS datasets were uncompressed and were re-gridded onto a monthly 1° X 1° standard World Geodetic System grid for use creating the gridded output, the grid has 180° of Latitude and 360° of Longitude with 0° centred on the Greenwich Meridian. Reformatting the datasets considerably reduced the file size to a more manageable ~50 GB. The scripts used to download these SST and SSS monthly datasets are fully automated, meaning that when new data are added to these datasets it can be easily incorporated into future versions of OceanSODA-UNEXE.

For each target variables (TA or DIC), the selected optimal SST and SSS gridded datasets and gridded WOA datasets were used as inputs to the respective TA and DIC algorithms. The algorithms were applied to the datasets in the two outflow regions where the SST and SSS dataset overlapped in time for the published environmental limits of each algorithm. The monthly SST and SSS inputs as well as the calculated monthly output variable (TA or DIC) are saved to a netCDF4 file as separate variables on the grid described above with dimensions of 768 by 180 by 360. The time dimension is the number of months since January 1957. It is important to note that TA and DIC datasets (even if for the same region) can span different temporal domains and so should be treated as separate datasets. These differences are due to differences in the temporal range of the input data used to calculate them.

The oceanic carbon system equations (Millero, 2000) can be solved computationally (Lewis et al., 1998), using numerical software packages (Orr et al., 2015) and the established CO2SYS software package is now available in Python as PyCO2SYS (Humphreys et al., 2022). Where there was temporal overlap between TA and DIC in each region, the remaining oceanic carbonate system variables ($pCO_2$ and pH) were calculated using PyCO2SYS v1.71 (Humphreys et al., 2022). A number of other output variables were also calculated including $fCO_2$, the carbonate ion content ($CO_3^{-2}$), the bicarbonate ion content ($HCO_3^-$), the hydrogen ion content ($H^+$) and the calcite ($\Omega$ Calcite) and aragonite ($\Omega$ Aragonite) saturation states. PyCO2SYS was run with the selected optimal dataset (SST, SSS and WOA datasets where applicable, with the same data used for input and output conditions) at the surface (0 m depth) with the carbonic acid dissociation constants of Mehrbach (1973) refitted by Dickson and Millero (1987) and the hydrogen sulphate dissociation constant of Dickson (1990). The full carbonate system is calculated twice, once using the optimal SST and SSS datasets that were used to calculate TA then again using those used to calculate DIC. Consequently each netCDF file contains a complete set of carbonate system outputs that are unique to that specific input dataset.

Whilst some of the SST and SSS products used to create the gridded datasets contain only satellite observed SST at a specified depth, some of the re-analysis SST and SSS products also incorporate surface ocean measurements from ships and buoys which were made at differing depths. The TA and DIC data used to generate the MDB comes from the top 10 m of the ocean and has not been adjusted for potential concentrations gradients in the surface ocean (Land et al., 2023). For these reasons, OceanSODA-UNEXE is deemed relevant in the surface 10 m of the ocean.

## 2.5 Uncertainties in gridded monthly time series product

As stated in section 2.3, the assessed combined standard uncertainties for TA and DIC are single fixed values provided as regionally-static non-spatially varying fields in the netCDF files. Uncertainty information for SST and SSS are taken from the literature references and are provided as spatially varying fields in the netCDF. Uncertainties in TA, DIC, SST and SSS are provided as optional uncertainty input arguments in PyCO2SYS to generate uncertainties for pH, $pCO_2$, $\Omega$ Calcite and $\Omega$

Aragonite. PyCO2SYS uses a forward finite difference approach to calculate the derivatives needed to propagate uncertainties for all these output variables. This produces a spatially varying combined uncertainty budget for each remaining carbonate system parameter. The spatially varying uncertainties in SST, SSS, pH, $p$CO$_2$, $\Omega$ Calcite and $\Omega$ Aragonite are all provided in the respective netCDF files.

## 3 Results

### 3.1 Algorithm evaluation


The algorithm evaluation RMSDe values in the Amazon and Congo for TA and DIC are shown in Figure 1. From these RMSDe values we selected the "optimal algorithms" for both regions as the algorithm combination with the lowest RMSDe. The optimal algorithms were then used to produce OceanSODA-UNEXE. In the Amazon the optimal algorithm to generate the TA data was (Cai et al., 2010) with ESACCI SST and ESACCI SSS (Table 1). To generate DIC for the Amazon the best

algorithm was (Lefèvre et al., 2010) using OISST SST and ESACCI SSS.  In the Congo, the optimal algorithm to generate the TA data was Lee et al. (2006) with CORA SST and ISAS SSS (Table 2). For DIC in the Congo the optimal algorithm was (Lee et al., 2000) with ESACCI SST and CORA SSS.


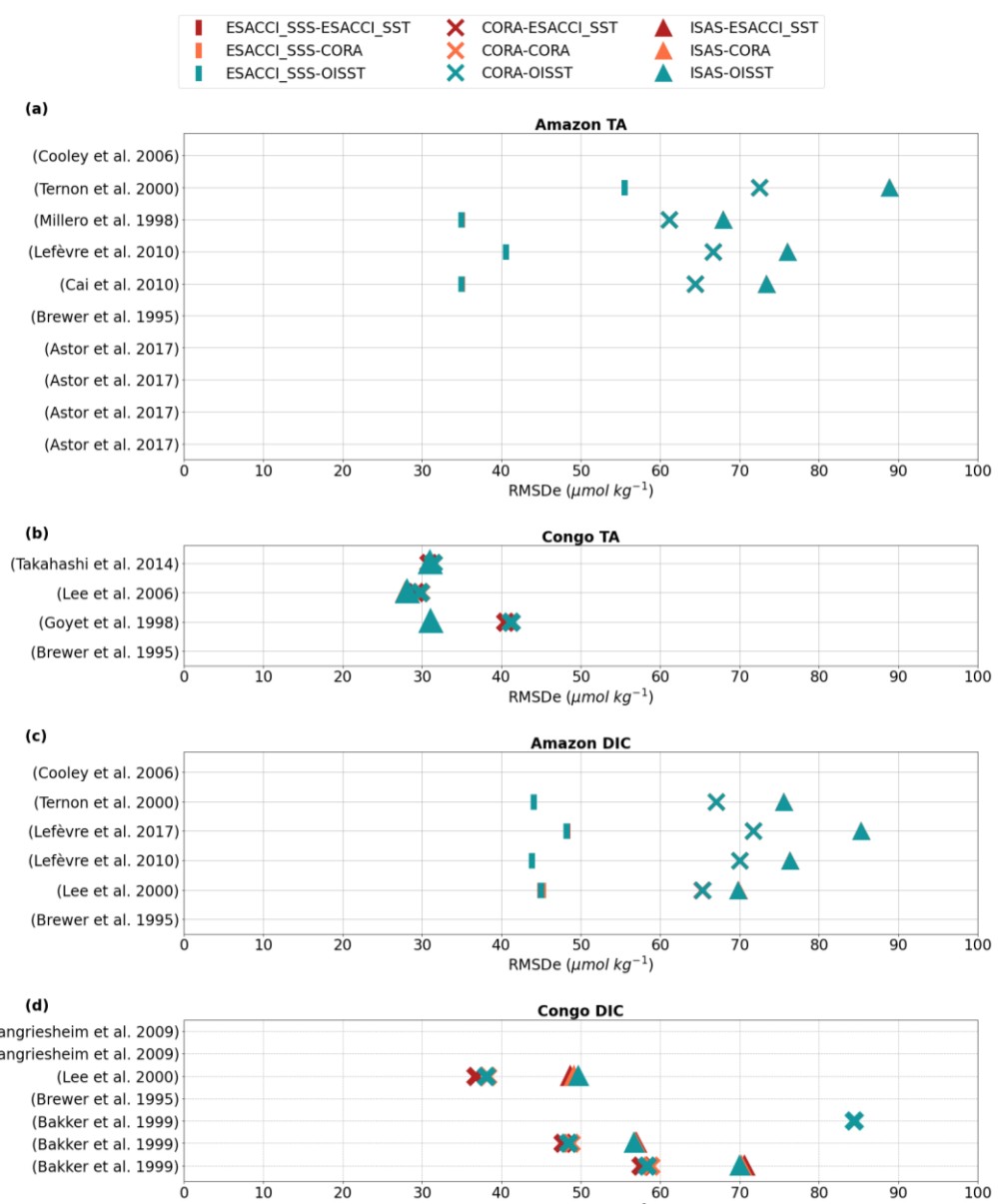

**Figure 1: Results of the algorithm evaluation, DIC and TA algorithms for the Amazon and Congo regions are compared by RMSDe. Each of the valid algorithms is listed by literature reference on the y axis, all of which correspond to algorithms in Table S1. Each SST and SSS dataset input configurations for which algorithms were evaluated are shown for each algorithm, with each SSS dataset given by a different symbol and each SST dataset given as a different colour in the legend. Entries are blank where there was no RMSD value in the literature, there were < 30 matchups or where the assessed RMSDe value was >100 µmol kg⁻¹.**


Table 1: Summary table of the best combination of algorithm and SST/SSS datasets (aka the "optimal algorithms") in the Amazon River domain. These optimal algorithms were determined by selecting the algorithm with the lowest RMSDe where there was at least 8 years of data.

| | Algorithm | SST dataset | SSS dataset | Years of overlap | Number of matchups (n) for weighted statistics | RMSD (µmol kg$^{-1}$) | wBias (µmol kg$^{-1}$) | wRMSD (µmol kg$^{-1}$) | RMSDe (µmol kg$^{-1}$) | Combined standard uncertainty (µmol kg$^{-1}$) |
|---|---|---|---|---|---|---|---|---|---|---|
| Optimal TA algorithm | (Cai et al., 2010) | ESACCI SST | ESACCI SSS | 10 years (2010 - 2020) | 82 | 50.92 | 7.83 | 34.97 | 34.97 | 35.84 |
| Optimal DIC algorithm | (Lefèvre et al., 2010) | OISST | ESACCI SSS | 10 years (2010 - 2020) | 70 | 53.33 | -5.98 | 43.83 | 43.83 | 44.23 |

Table 2: Summary table of the best combination of algorithm and SST/SSS datasets (aka the "optimal algorithms") in the Congo River domain. These optimal algorithms were determined by selecting the algorithm with the lowest RMSDe where there was at least 8 years of data.

| | Algorithm | SST dataset | SSS dataset | Years of overlap | Number of matchups (n) for weighted statistics | RMSD (µmol kg$^{-1}$) | wBias (µmol kg$^{-1}$) | wRMSD (µmol kg$^{-1}$) | RMSDe (µmol kg$^{-1}$) | Combined standard uncertainty (µmol kg$^{-1}$) |
|---|---|---|---|---|---|---|---|---|---|---|
| Optimal TA algorithm | (Lee et al., 2006) | CORA | ISAS | 14 years (2002 - 2016) | 102 | 27.33 | 6.00 | 27.91 | 27.91 | 28.54 |
| Optimal DIC algorithm | (Lee et al., 2000) | ESACCI SST | CORA | 14 years (2002 - 2016) | 77 | 37.37 | -16.64 | 36.79 | 36.79 | 40.37 |


### 3.2 Dataset output

#### 3.2.1 Amazon dataset

The primary output datasets are the monthly gridded carbonate system variables (DIC and TA). The seasonal progression of both these variables is shown in the Amazon outflow region in Figures 2 and 3. There is a large range of DIC and TA in this region, with high values (DIC ~2000 µmol kg$^{-1}$ and TA ~2350 µmol kg$^{-1}$) in the open ocean. The lowest values (DIC and TA ~1400 µmol kg$^{-1}$) are located at the mouth of the river. Intermediate values occur where the plume and ocean water are mixing. The plume extends to the Northwest from the Amazon River mouth (Coles et al., 2013); this is especially visible from April to September (Figures 2 and 3 plots b and c), less so from October to March (plots a and d), demonstrating the seasonal variability. The region of influence of the Amazon Plume extends into the Caribbean between April and June (Figure 2b and 3b), a feature reported elsewhere (Chérubin and Richardson, 2007). From July to September (Figure 2c and 3c) another large outflow region occurs just off the coast of Venezuela at the mouth of the Orinoco River (Hu et al., 2004). The Orinoco plume is much less prominent in the other seasons. The proximity of the Orinoco River to the Caribbean suggests that the Orinoco plume often reaches the islands between July and September (López et al., 2013). There are some differences between the structure of the average TA and DIC plumes which may indicate active processes in the plume, for example gas exchange which only affects DIC or biological production which affects TA and DIC at different rates. Annual and seasonal data averages over the region shown in Figures 2 and 3 are provided in Table S3 for DIC, TA, pH, $p$CO$_2$, Ω Calcite and Ω Aragonite.

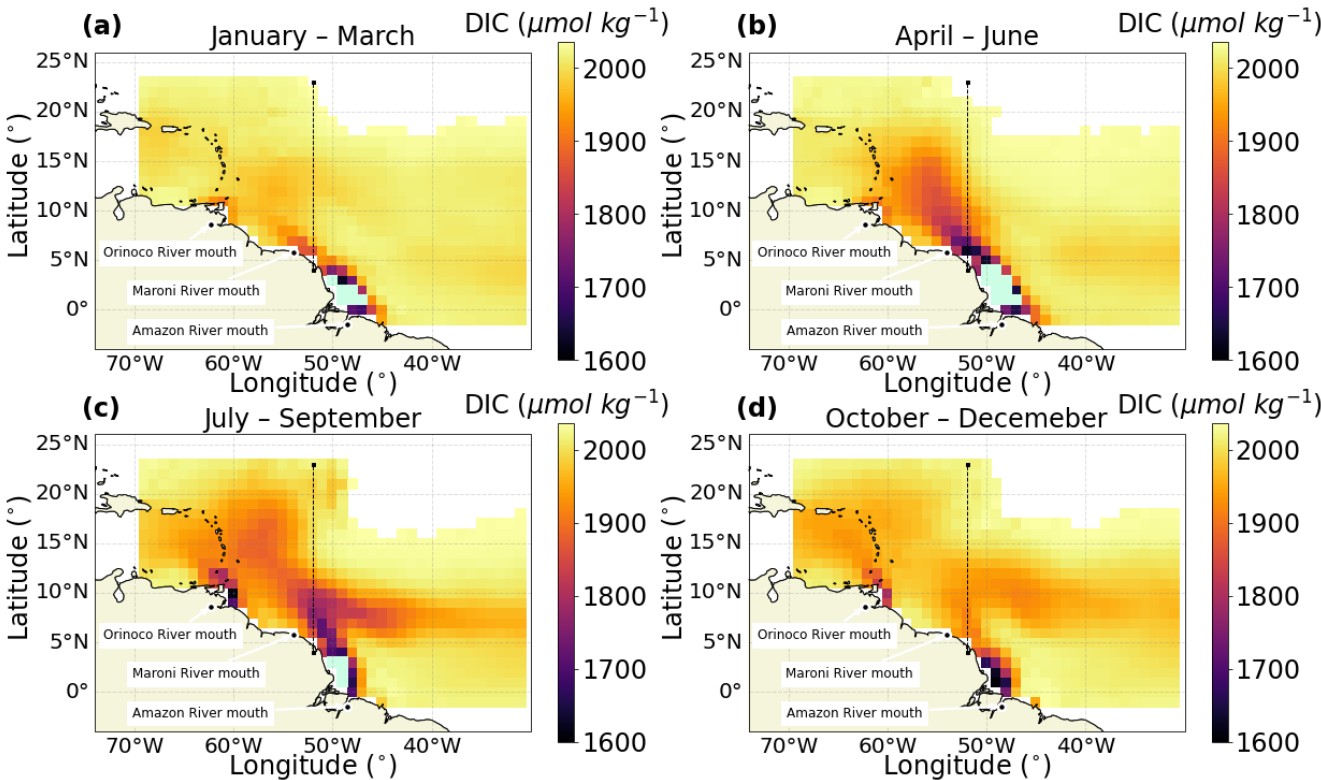

**Figure 2: Seasonally averaged DIC for the Amazon plume region in (a) January to March (b) April to June (c) July to September (d) October to December. Land outlines are shown in beige. Ocean regions out of bounds or where there was no algorithm output are left white. Algorithm data below 1600 µmolkg[-1] at the river outflows is shown in mint green. The mouths of the Amazon, Orinoco and Maroni Rivers are labelled. The 52° W meridional section used for the Hovmöller plot used in Figure 4 is indicated as a black dashed line.**

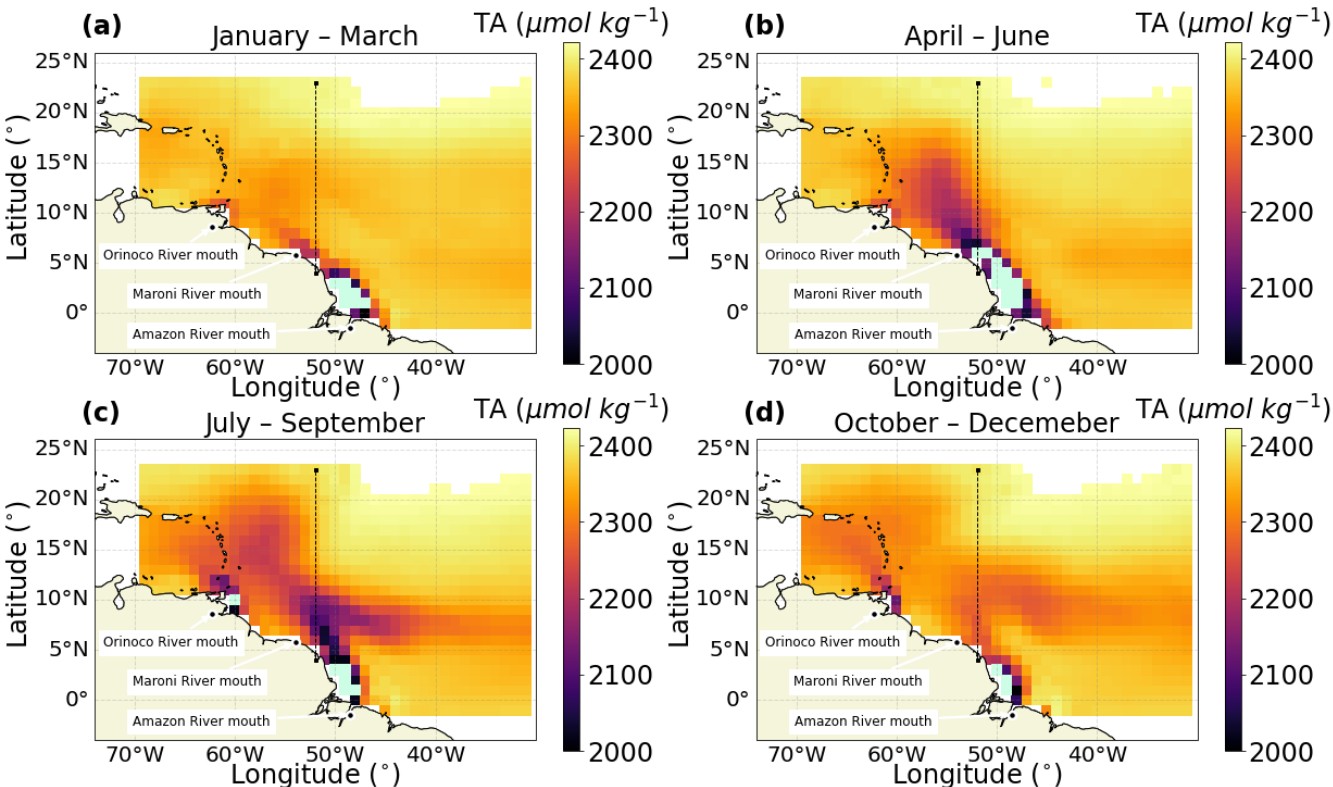

**Figure 3: Seasonally averaged TA for the Amazon plume region in (a) January to March (b) April to June (c) July to September (d) October to December. Land outlines are shown in beige. Ocean regions out of bounds or where there was no algorithm output are left white. Algorithm data below 2000 µmolkg$^{-1}$ at the river outflows is shown in mint green. Orinoco and Maroni Rivers are labelled. The 52° W meridional section used for the Hovmöller plot used in Figure 4 is indicated as a black dashed line.**

The temporal aspect of the dataset is presented in Figure 4 as a meridional Hovmöller plot at 52° W (marked on Figures 2 and 3) which broadly cuts through the central plume outflow at 4° N, the mixing of the plume up to 10° N and the open ocean Atlantic water >10° N. The lowest DIC values (~1400 µmol kg-1) are found in the plume at 4° N, there is regular seasonality with lower values in the October to December and higher values in April to June (Figure 4a). There does not appear to be much interannual variability in the magnitude of the DIC plume in the June to August period, although 2010 and 2013 appear to be weaker (Figure 4a). The maximum northerly extent of the plume varies from year to year, with higher 315 DIC found at 15° N in the May to August period of 2011. The opposite is true in other years, for example the plume does not extend as far northwards in 2013 (Figure 4a). TA shows much more interannual variability (Figure 4b). The minimum plume values do not always occur at the same latitude or at the same time. In most years the minimum plume values are as low as 1750 µmol kg$^{-1}$ but this was not the case in 2016. The northward extent of the plume is variable, in some years the 320 plume extended much further North, e.g. June to July 2011, and in other years the plume influence was not as detectable

further North, e.g. May to July 2016. The period of time where the plume dominates the region is also variable; in some years the peak plume intensity which begins in May lasts through to September while in other years it is already declining in July.

As $p\mathrm{CO_2}$ and pH are derived from DIC and TA, the behaviour of these variables mirrors that of TA and DIC (Figure 4c and 4d). The river plume always has above neutral pH and occasionally extends far to the north, e.g. June to July 2011. There are low $p\mathrm{CO_2}$ values (<200 µatm) in the plume which are much lower than some of the values observed in the plume (Lefèvre et al., 2017). Calcite and aragonite saturation states have the expected magnitude in open ocean regions and are lower in the plume, with $\Omega$ Calcite levels ~3 in the plume.


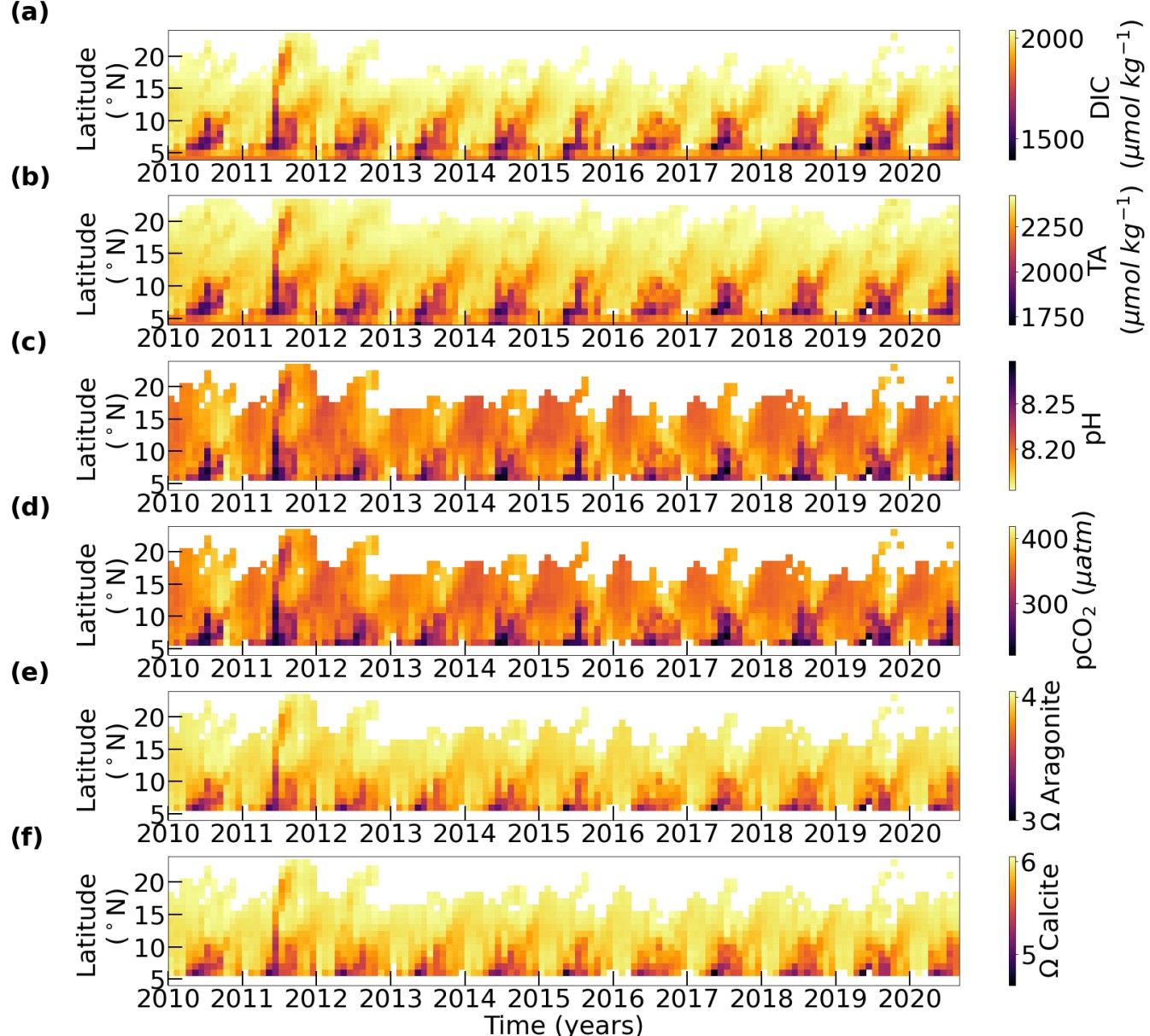

**Figure 4: Hovmöller plots in the Amazon outflow region for (a) DIC, (b) TA, (c) pH on the free scale, (d) $p$CO$_2$, (e) Ω Aragonite and (f) Ω Calcite. The plots span the 52° W meridian from 4 to 24° N. The plots cover the temporal overlap period of the TA and DIC datasets 2010- 2020.**


The gridded data can also be spatially averaged to show the evolution of each of the carbonate system variables in time within the plume (defined as SSS <35 (Grodsky et al., 2014)), outside the plume (SSS>35) and over the whole region (Figures 5). The mean DIC calculated across whole region shows a very consistent seasonal pattern, with higher DIC in

December to February (~2000 μmol kg$^{-1}$) and lower DIC in June to August (~1970 μmol kg$^{-1}$). The mean values within the plume are much lower; the lowest DIC values (~1700 μmol kg$^{-1}$) are seen in December to February and the highest values (~1900 μmol kg$^{-1}$) are during September and October. The mean non-plume values show much smaller seasonal variability. In the plume, outside of the plume and the whole region, surface DIC is very similar year to year. The mean DIC within the plume reaches a minimum in 2013 which was much lower than all other years. Non-plume DIC values increased by 0.49 μmol kg$^{-1}$ per year. TA across the whole region is consistent inter annually, with a seasonal maximum in December to February (~2360 μmol kg$^{-1}$) and minimum in July to August (~2330 μmol kg$^{-1}$), while TA within the plume shows a seasonal maximum in October (~2250 μmol kg$^{-1}$) and minimum in April to May (~2150 μmol kg$^{-1}$). The annual minimum within the plume varies from December to March, while the maximum consistently occurs in October and November, with considerable interannual variability of the timing of the annual minimum, with much lower TA in the December to February periods of 2012/2013 and 2017/2018 than over the same period in other years. Non-plume TA values increased by 0.76 μmol kg$^{-1}$ per year. It should be noted that the whole region is out of phase with the plume, this is because whilst the plume is more dilute this effect is outweighed by the larger plume between May and August.

For pH there is a weak seasonal trend in the overall region and outside the plume. There is a lot more variability in the plume, with the highest pH values (~8.225) occurring in December and the lowest values (~8.2) in September. The best SST and SSS dataset from the DIC algorithm predict that the pH will be >0.1 higher than the best SST and SSS dataset from the TA algorithm between January and March. The two estimates agree within <0.1pH units for the other 9 months of the year. For $pCO_2$, also computed from TA and DIC, mean values in the non-plume region and over the whole region were very stable, which is consistent with the minimal season variability seen in oligotrophic oceans. Whereas there was considerable variability in plume $pCO_2$, with average values of ~325 μatm seen in the plume in March and values of ~350 μatm in August. The differences between the $pCO_2$ calculated with the different SST and SSS datasets was very small, with differences <5 μatm most of the year. The average calcite and aragonite saturation states are in the typical range for seawater and are much greater than the critical thermodynamic threshold for calcification with a value of 1 (Waldbusser et al., 2016). It should be noted that an aragonite saturation state threshold of 3 has also been recommended for warm water corals (Guinotte et al., 2003), other studies have since shown that warm water corals can still adapt and survive in these conditions (Enochs et al., 2020;Uthicke et al., 2014). There are periods in the data such as at the start of 2019 where the aragonite saturation state falls below 3. The mean saturation states within the plume are lower than those found in the wider region.

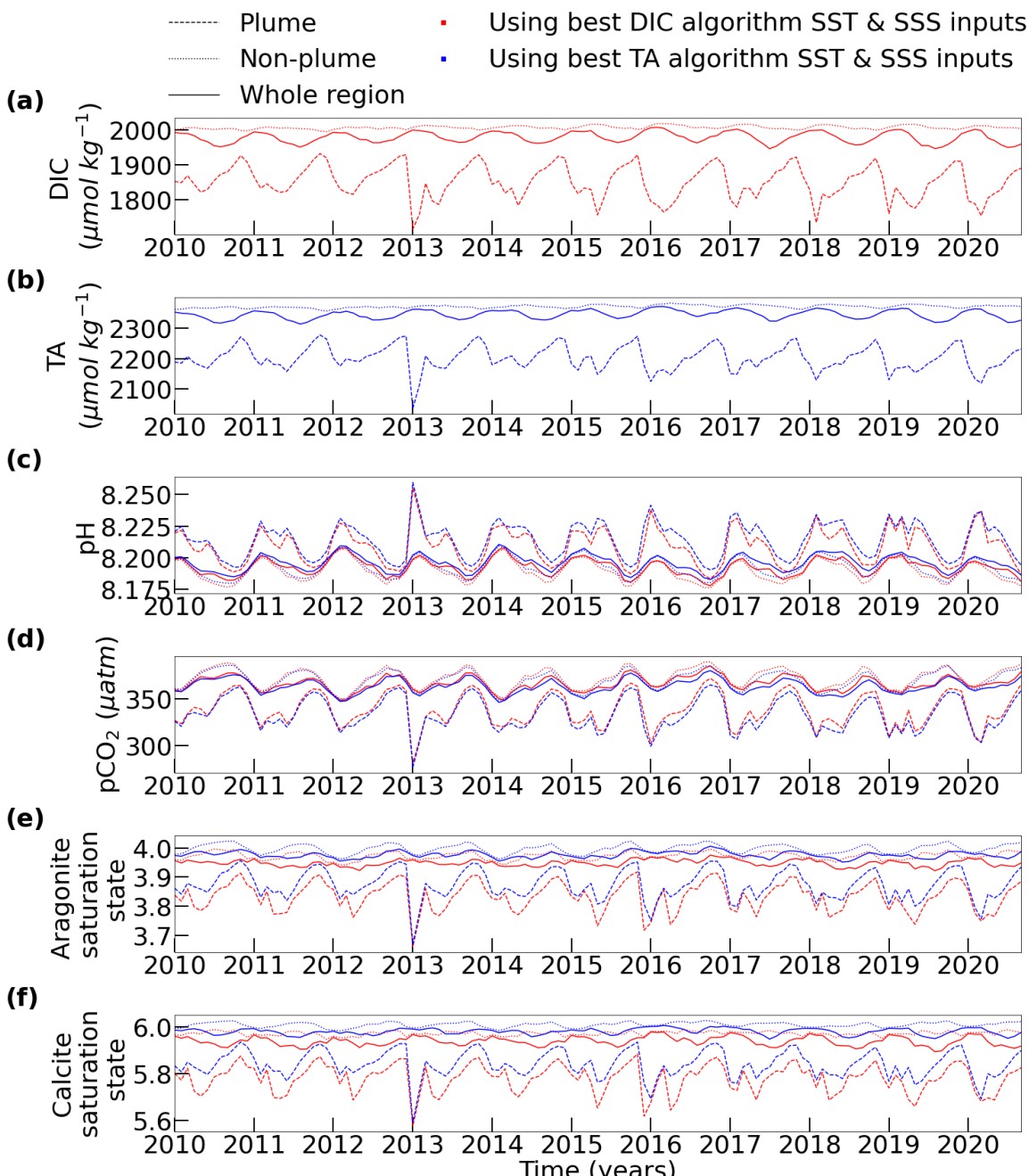

**Figure 5:** Timeseries of spatially averaged (a) DIC, (b) TA, (c) pH, (d) $p$CO$_2$, (e) Ω Aragonite and (f) Ω Calcite in the Amazon region. The plots spans the temporal overlap period of the TA and DIC datasets 2010- 2020. Data are averaged across the whole region (solid line) as well as in the plume defined as S<35 (dashed line) and outside of the plume S>35 (dotted line). The line colour corresponds to variables that were calculated with SST and SSS datasets selected during the DIC algorithm evaluation (red) and the TA algorithm evaluation (blue).

### 3.2.1 Congo dataset

In the Congo outflow region, there is a strong seasonal variation in DIC (Figure 6). From July to September (Figure 6c) there are two regions of low DIC, one directly at the outflow point of the Congo River and another at the outflow point of the Niger River Delta. The outflows of both of these river dominated regions extend directly west, aligned to the South Equatorial Current in the eastern Atlantic (Hopkins et al., 2013). In October to December (Figure 6d) the spatial influence of the Congo outflow is smaller possibly due water masses moving northwards along the coast. Into January to March  (Figure

6a) there is a region of low DIC along the whole coast, representing the intensification of the Niger and Congo discharges at this time of year (Chao et al., 2015). In April to June (Figure 6b) the coastal water mass separates becoming two distinct outflows, the outflows reach their greatest westward extent covering almost the whole domain.

    Similar trends are seen in TA (Figure 7) with two distinct outflow regions in July to September (Figure 7c). The outflows

begin to intensify in October to December (Figure 7d). The lowest TA values are observed in January to March to the east of the Niger Delta outflow (Figure 7a). Unlike in Figure 6a, the outflows remain distinct in Figure 7a. Between April and June the outflow reaches its maximum spatial extend as it flows out to the west where it is has a detectable impact across the region.


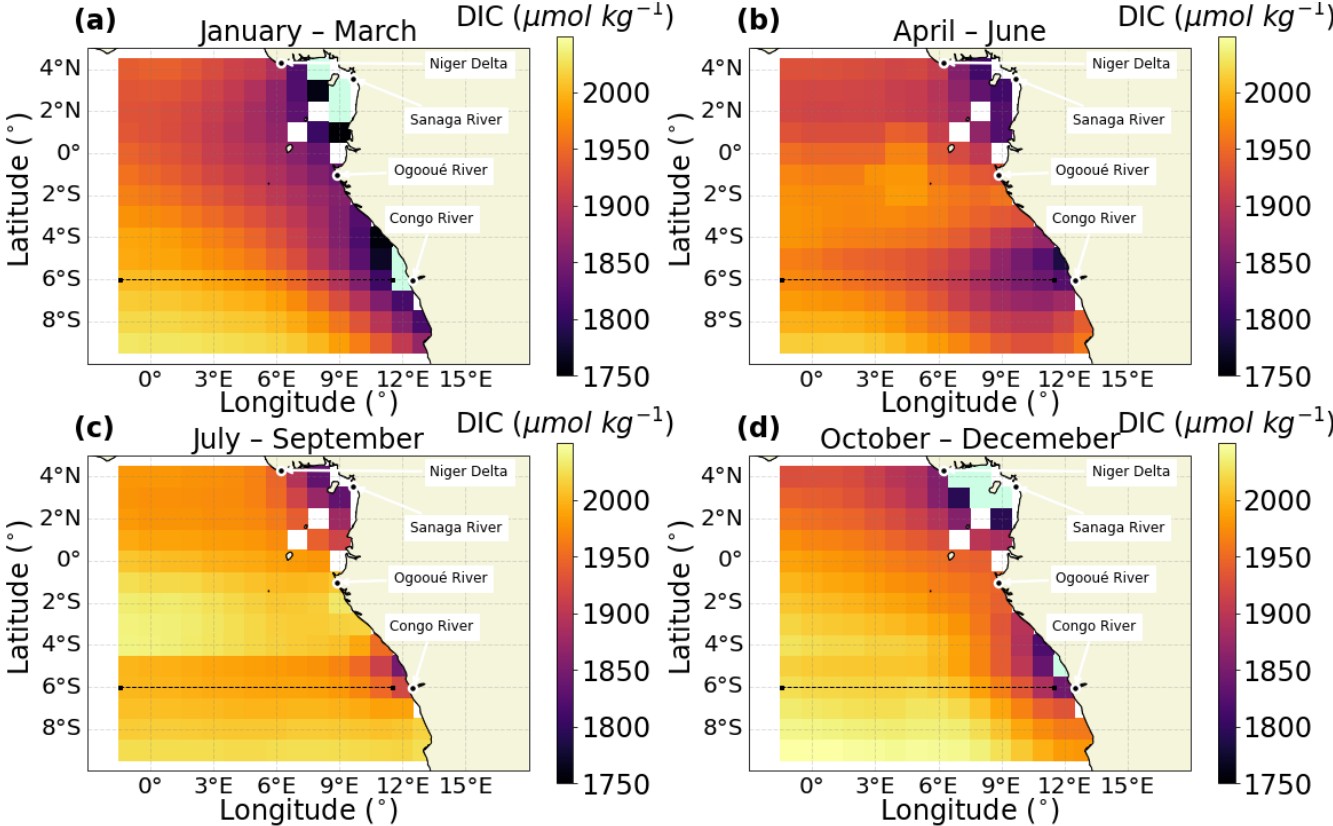

**Figure 6:** **Seasonally averaged DIC for the Congo outflow region in (a) January to March (b) April to June (c) July to September (d) October to December. Land outlines are shown in beige. Ocean regions out of bounds or where there was no algorithm output are left white. Algorithm data below 1750 µmolkg$^{-1}$ at the river outflows is shown in mint green. The Niger River Delta and the mouths of the Congo, Ogooué and Sanaga Rivers are labelled. The 6° S meridional section used for the Hovmöller plot used in Figure 8 is indicated a bold dashed line.**


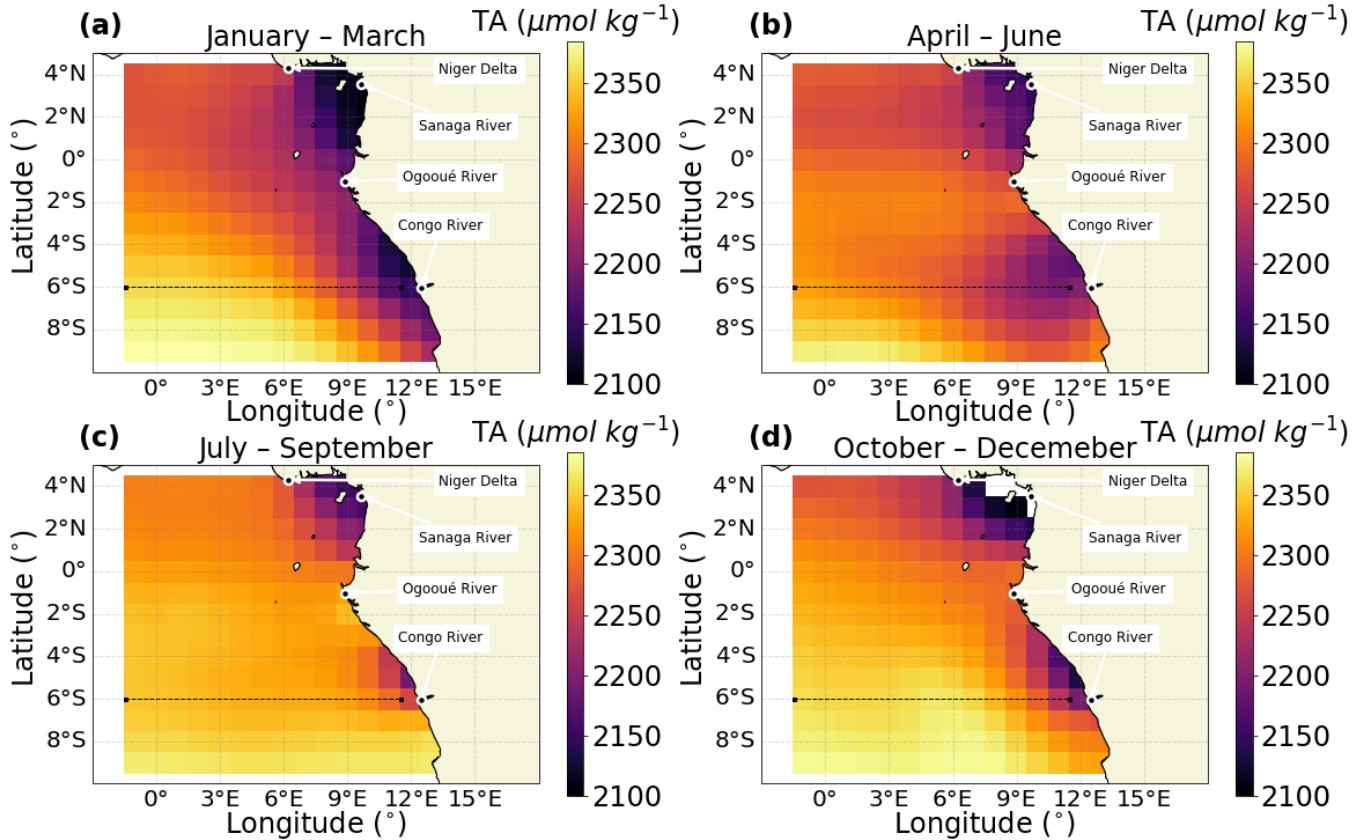

Figure 7:  Seasonally averaged TA for the Congo outflow region in (a) January to March (b) April to June (c) July to September (d) October to December Land outlines are shown in beige. Ocean regions out of bounds or where there was no algorithm output are left white. The Niger River Delta and the mouths of the Congo, Ogooué and Sanaga Rivers are labelled. The 6° S meridional section used for the Hovmöller plot used in Figure 8 is indicated a bold dashed line.

A zonal section at 6° S was used to construct a Hovmöller plot of variables in the Congo (Figure 8). This latitude is centred across the outflow of the Congo River. Unlike in the Amazon, much of the plot region is masked in between May and July as the published algorithms were not valid for full environmental conditions experienced by the region.

The DIC plot (Figure 8a) shows that the outflow is low in DIC (~1800 μmol kg$^{-1}$) and the open ocean is higher (~2050 μmol kg$^{-1}$). The highest values are found in the January to March period, consistent with Figure 6a. The outflow is detectable over the widest area in the March to June period in all years.  Whilst there is one period of intense outflow each year there are is also some indication that a weaker outflow is intermittently detectable in the data (shown as vertical streaks in Figure 8a). TA (Figure 8b) shows an almost identical pattern to DIC. The lowest TA values (~2100 μmol kg$^{-1}$) were observed between January and March and slightly higher values (~2200 μmol kg$^{-1}$) were seen when the outflow extended further West between April and June.

Higher pH values (~8.3) were observed in the outflow compared to the open ocean (~8.2) (Figure 8c. Previous studies have
measured low pH values in the main body of the river and its tributaries (Wang et al., 2013;Bouillon et al., 2014), the higher
pH values at the mouth of the river may be due to complex carbonate speciation. The minimum $p$CO$_2$ values are very low
(~200 μatm) in the inner part of the outflow (Figure 8d), while values in the outer part of the outflow are closer to the
expected $p$CO$_2$ values around 350 μatm. (da Cunha and Buitenhuis, 2013). Mirroring the trends in pH, the calcite and
aragonite saturation states (Figure 8e and 8f) are higher in the outflow relative to the open ocean.  Anomalous pH, $p$CO$_2$ and
the calcite and aragonite saturation states values in December 2003 are likely associated with the heavy rainfall over south-
central Africa (Kadomura, 2005).

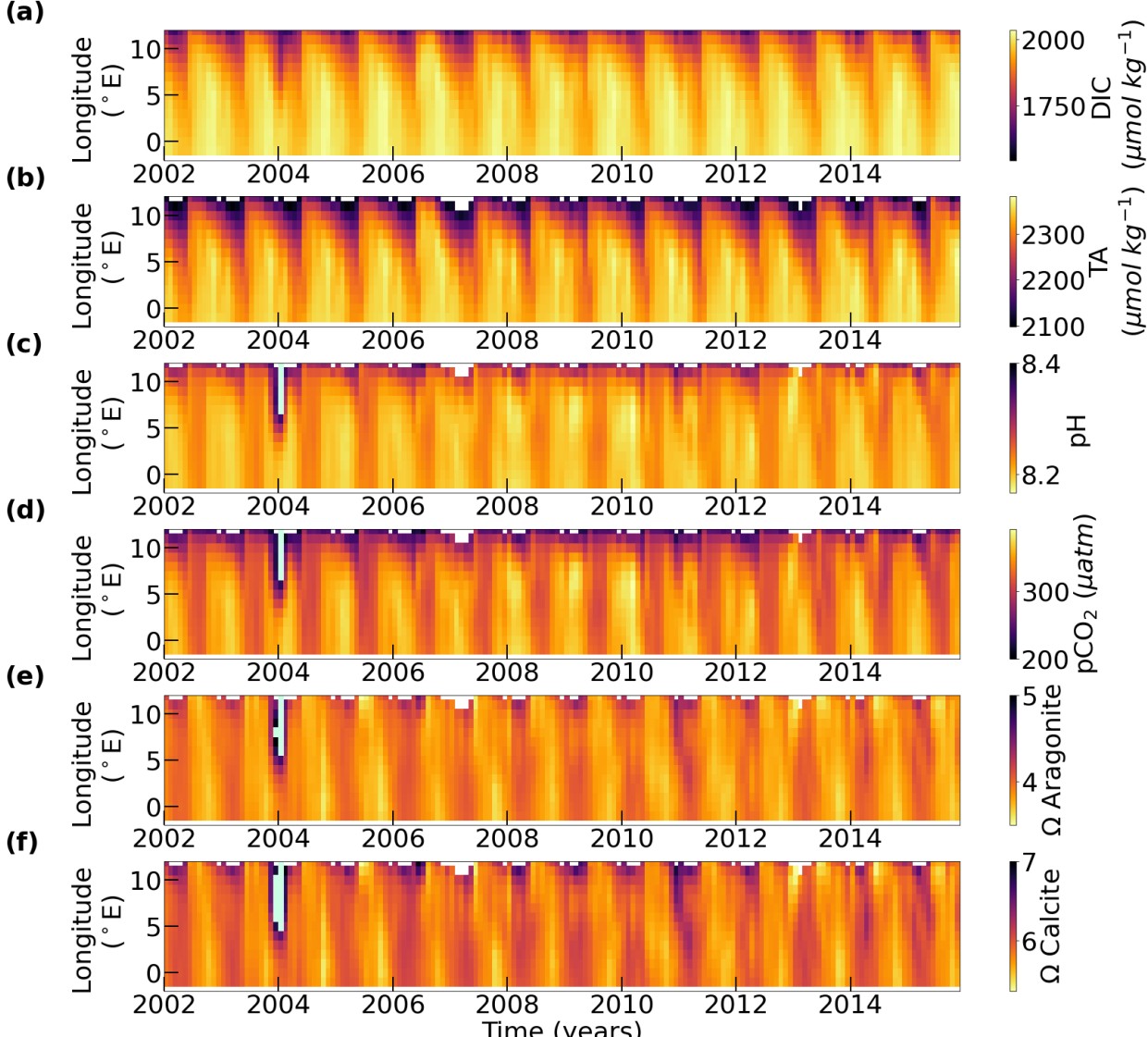

Figure 8: Hovmöller plots of (a) DIC, (b) TA, (c) pH, (d) $p$CO₂, (e) Ω Aragonite and (f) Ω Calcite for the Congo outflow region. The plots are centred on the 6° S slice that spans 2° W to 12° E. pH > 8.4, $p$CO₂ data < 200µatm, Ω Calcite>5 and Ω Calcite>7 in the river outflow in December 2003 are shown in mint green. The plots spans the temporal overlap period of the TA and DIC datasets 2002- 2016 and thus the period for which the rest of the carbonate system was generated.

The mean DIC and TA across the whole Congo region, in the outflow and out of the outflow are very consistent year to year (Figure 9a and 9b). The yearly TA and DIC minima occur at approximately the same time and are the same magnitude most years. DIC and TA values are always higher in the non-outflow region than in the outflow. This consistency is also seen in the propagated variables pH (Figure 9c), $p$CO₂ (Figure 9d), and the aragonite and calcite saturation states (Figure 9e and 9f) all of which show very minor differences between the outflow and non-outflow regions.

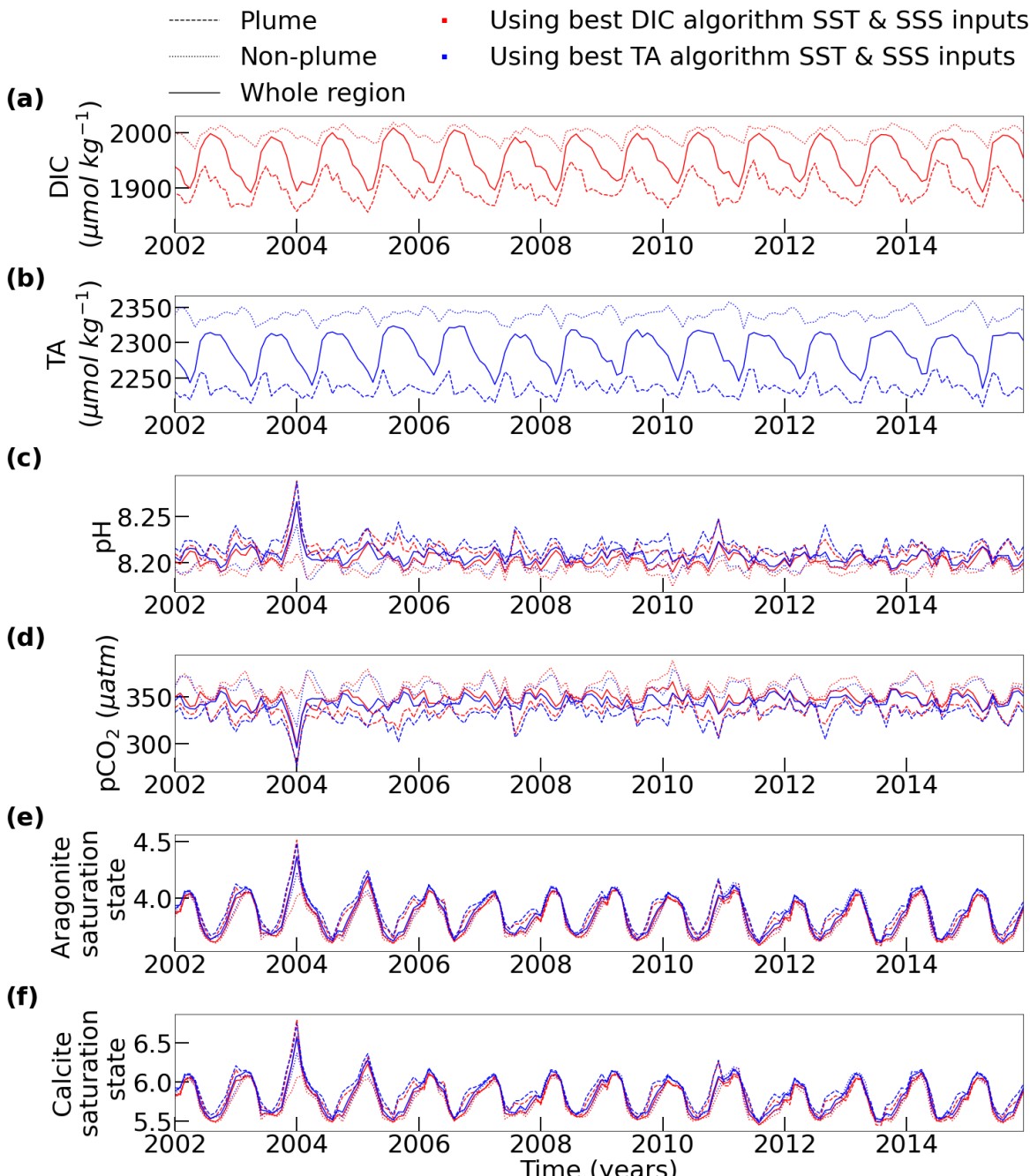

**Figure 9: Timeseries of spatially averaged (a) DIC, (b) TA, (c) pH, (d) $p$CO₂, (e) Ω Aragonite and (f) Ω Calcite in the Congo region.** The plots spans the temporal overlap period of the TA and DIC datasets 2002- 2016. Data are averaged across the whole region (solid line) as well as in the outflow defined as S<35 (dashed line) and outside of the outflow S>35 (dotted line). The line colour corresponds to variables that were calculated with SST and SSS datasets selected during the DIC algorithm evaluation (red) and the TA algorithm evaluation (blue).

### 3.4 Uncertainty assessment

The TA and DIC combined standard uncertainties from the algorithm evaluation are shown in Table 1 and 2. The combined standard uncertainties for the optimal algorithms used to generate the time series dataset are reproduced in Table 3. The combined standard uncertainties are the first Type A uncertainty evaluation, which can be thought of as a "top down" estimate. The spatially averaged uncertainties of the remaining carbonate system variables using this top down estimate are shown in both outflow regions (Table 3). The uncertainties of the remaining carbonate system variables appear to be weakly

dependent on the choice of SST and SSS datasets used to calculate them in PyCO2SYS (Table 3). Uncertainties are provided as absolute values rather than as percentages because of the steep gradient in values across the plume, but if required the uncertainty values can be expressed as a percentage using the minimum, maximum and mean values from table S4.

The second Type A uncertainty evaluation (using the algorithm RMSD from the literature and the uncertainties in the inputs)

can be thought of as a "bottom up" estimate. For this bottom up estimate, the average combined standard uncertainties in TA and DIC are smaller for TA (8.52) and DIC (16.59) in the Amazon and in the Congo for TA (17.23) and DIC (14.25). These values are much lower than from the top down uncertainty evaluation as this uncertainty evaluation does not fully account for spatial and depth variability (by assuming that the literature RMSD values perfectly capture the variability), these values also do not account for measurement variability. The bottom up evaluation does account for uncertainty in the input SST and

SSS datasets suggesting spatial and depth variabilities and differences between the in situ data (used to evaluate the uncertainties) are likely dominating the uncertainty budget

As the top down uncertainty evaluation is more robust it is the preferred uncertainty estimate. By comparing the uncertainties with the natural variability in TA and DIC (Table 4) it is clear that the dataset uncertainty is less than the

natural variability data in both riverine regions. Whereas, the propagated uncertainties for $p\mathrm{CO_2}$ and pH are larger than the natural variability, this is only because the $p\mathrm{CO_2}$ and pH data generated in OceanSODA-UNEXE has full spatio temporal coverage including in the high variability plume compared to the $p\mathrm{CO_2}$ and pH *in situ* data in the MDB.

Table 3: Output variable uncertainties in the two regions of OceanSODA-UNEXE. TA and DIC uncertainties are the

combined standard uncertainties from the algorithm evaluation. The uncertainty for the remaining carbonate system variables are the average of the spatially varying propagated uncertainties for each of those variables calculated with the SST and SSS pair from either the TA or DIC algorithm. Note that due to the logarithmic nature of pH, uncertainty is calculated as 1σ in pH units and also in pH units where variability was determined in H$^+$ as $-\log_{10}(\overline{H^+}) + \log_{10}(\overline{H^+} + 1\sigma\, H^+)$.


| | | | Using SSS and SST from TA algorithm | | | | | Using SSS and SST from DIC algorithm | | | | |
|---|---|---|---|---|---|---|---|---|---|---|---|---|
| Variable | TA | DIC | $p$CO$_2$ | pH converted from H$^+$ | pH | ΩCal | ΩArag | $p$CO$_2$ | pH converted from H$^+$ | pH | ΩCal | ΩArag |
| Combined uncertainty in the Amazon | 34.74 | 44.34 | 85.19 | 0.08 relative to a pH of 8.20 | 0.07 relative to a pH of 8.19 | 0.91 | 0.61 | 86.17 | 0.08 relative to a pH of 8.19 | 0.08 relative to a pH of 8.19 | 0.91 | 0.60 |
| Combined uncertainty in the Congo | 28.54 | 33.25 | 73.14 | 0.07 relative to a pH of 8.21 | 0.08 relative to a mean pH of 8.21 | 0.79 | 0.52 | 74.02 | 0.07 relative to a pH of 8.20 | 0.08 relative to a mean pH of 8.20 | 0.79 | 0.52 |

Table 4: *In situ* measurement variabilities for TA, DIC, pH and $p$CO$_2$ are calculated as the standard deviation of the reference output (MDB).For the Amazon ESACCI SST and ESACCI SSS were the combination used with the MDB and for the Congo this was CORA SST and ISAS SSS. Note that due to the logarithmic nature of pH, uncertainty is calculated as $1\sigma$ in pH units and also in pH units where variability was determined in H$^+$ as $-\log_{10}(\overline{H^+}) + \log_{10}(\overline{H^+} + 1\sigma\ H^+)$.

| Variable | TA | DIC | $p$CO$_2$ | pH converted from H$^+$ | pH | ΩCal | ΩArag |
|---|---|---|---|---|---|---|---|
| *In situ* measurement variability (1σ) in the Amazon from the MDB | 104.47 | 102.96 | 25.86 | 0.037 relative to a pH of 8.07 | 0.043 relative to a mean pH of 8.07 | – | – |
| *In situ* measurement variability (1σ) in the Congo from the MDB | 46.13 | 51.71 | 31.36 | 0.003 relative to a pH of 8.03 | 0.003 relative to a mean pH of 8.03 | – | – |

**4 Discussion**


The algorithm evaluation (Figure 1) demonstrated that the choice of input SSS dataset to the TA and DIC algorithms makes the biggest difference in reducing the RMSDe in TA and DIC. SST tends to be a secondary term in the majority of the algorithms so has less of a controlling effect. The choice of literature TA and DIC algorithm itself impacts RMSDe to a much smaller extent than the choice of input SSS datasets. The prominence of SSS terms in the majority of the algorithms

explains why the RMSDe is much more sensitive to the choice of SSS dataset compared to SST dataset, as SST is a secondary term in the majority of the algorithms. Whilst some algorithms did perform better than others the differences were so slight that it may not be that helpful to declare which of the algorithms are the best outright especially as this could change in the future with more data.

The requirement of having n=30 matchups to calculate the weighted statistics has a large impact on the choice of optimal algorithms. By including this stipulation, some of the more recent salinity input datasets are effectively de-selected as there are not enough contemporary measurements over their temporal range to allow their evaluation, particularly in the Congo region. For example, the SMAP satellite only launched in 2015 and there were not enough *in situ* matchups in either region to be able to fully assess the RSS-SMAP dataset. So using ISAS for salinity in the optimal TA algorithm in the Congo where

data are available from 2002 onwards) increases the number of matchups by a factor of twenty compared to using RSS-SMAP (Table 2). Given time, the number of matchups with the satellite only products will increase allowing them to be fully evaluated. Considering the better spatial coverage of the satellite only products, it is likely that in the near future the best algorithm combinations to generate updates to this product will all use satellite derived SSS and SST. The recent proliferation of certified reference materials when measuring TA (Dickson et al., 2003) and DIC (Dickson, 2001) should also

mean that newer *in situ* data will have lower uncertainties. The relatively low numbers of insitu data limits the potential use of neural network based approaches to estimate the carbonate system in these outflows and significantly more in situ data would be needed (for training and testing) before these approaches could likely be applied.

By comparing the two type A uncertainty estimates, the standard combined uncertainty estimate (accounting for

measurement uncertainty, spatial uncertainty, depth uncertainty and algorithm uncertainty, aka "top down") and the second type A uncertainty evaluation (propagated input uncertainties through the literature algorithm, aka "bottom up"); the contribution of different factors to the uncertainty can be dissected. The bottom up uncertainty estimates are much smaller than the top down uncertainty estimates reflecting the fact that less of the uncertainty has been considered in the bottom up uncertainty estimates. The difference between the top down and bottom up estimates can be mainly attributed to the spatial

and depth uncertainty which is accounted for by using the MDB in the algorithm evaluation. Land et al. (2019) note that reducing the uncertainty of the *in situ* measurements is just as important as all the remaining uncertainties in their full

methodology. Further reducing the standard combined uncertainty is challenging and would require either improvements in satellite SSS retrievals (Vinogradova et al., 2019) or improvements in how uncertainty is quantifying in the GLODAP data (a huge challenge give the different systems and protocols used by different laboratories) (Bockmon and Dickson, 2015).


The timeseries data demonstrates that there is low TA and DIC in the river outflows of both regions. The timeseries data clearly shows that the discharge of the outflows and their zone of influence changes seasonally. Our results are consistent with previous observations that the Amazon plume extent is smallest between January and March (Fournier et al., 2015) and largest in the between April and June where it expels low TA and DIC waters (Cooley and Yager, 2006). The Amazon

plume reaches the Caribbean, as previously shown by Hellweger and Gordon (2002). The timeseries data identify that in parts of the Amazon plume saturation states can drop below 3 (Table S4) which has implications for OA research, especially for researchers studying coral reefs. The non-plume DIC and TA trends of 0.49 and 0.76 μmol kg$^{-1}$ per year are consistent with the decadal increases in salinity normalised DIC (4.54 μmol kg$^{-1}$) and salinity normalised TA (7.39 μmol kg$^{-1}$) in the 2010s from the Bermuda Atlantic Time-series Study site (Bates and Johnson, 2020). The results identify that the discharge of

the Congo outflow is greatest between January and March and veers towards West Africa, which is consistent with work by Hopkins et al. (2013).

The timeseries data comprises over ten years of carbonate system data for two of the world's largest rivers by discharge. One immediate use for the timeseries data would be to assess the inorganic carbon flow of both of these rivers over time.

The timeseries data could also be used to investigate the impact of charges in land use within the river basin, e.g. deforestation impacts on river discharge and inorganic carbon flow into rivers (Bass et al., 2014). If the impact of land use changes were identifiable within the data then these approaches may prove a useful tool in monitoring the effectiveness of policy or management actions addressing climate change and biodiversity loss in the Amazon Basin. The timeseries data could also be used for ocean acidification research (Land et al., 2015), as these rivers discharge enough fresh water to

influence carbonate saturation states. The tropical reefs of the Caribbean are infrequently impacted by the Amazon plume (Chérubin and Richardson, 2007); the impact of the low pH waters on reef health is of great concern (Hoegh-Guldberg et al., 2007) and could be explored with the timeseries data. Ocean acidification has been shown to impact foraging behaviour in fish (Jiahuan et al., 2018) and sharks (Rosa et al., 2017), the timeseries data could be used in conjunction with GPS tracks of fish or marine mammals to see if the outflows alter foraging behaviour. The timeseries data could also be used to explore

CO$_2$ fluxes in the Amazon outflow building upon more recent estimates (Olivier et al., 2022;Ibánhez et al., 2016;Mu et al., 2021).

**5 Conclusions**

OceanSODA-UNEXE is a time series dataset of the carbonate system in the outflow regions of the Amazon and Congo rivers. Optimal TA and DIC data are generated with the optimal combination of published algorithms and input datasets determined by an exhaustive round robin inter-comparison evaluation. By using a specially designed matchup database for the algorithm evaluation, uncertainties due to spatial and depth variability in the *in situ* references have been minimised. TA, DIC, SST and SSS are used as inputs into PyCO2SYS to calculate the remaining carbonate system variables (pH and $pCO_2$). TA and DIC are provided with standard combined uncertainties from a Type A uncertainty evaluation whereas pH and $pCO_2$ are provided with propagated uncertainties from PyCO2SYS. The assessed uncertainties are lower than the natural variability within these regions and the main features of both river outflows are evident in all of the carbonate system variable outputs. Potential uses for these data could include evaluating the riverine carbon flux from the land into the ocean resulting from the Amazon and Congo rivers, or evaluating the extent that river-driven episodic changes in the carbonate system may be having on sensitive coral reefs that interact with the outflows.

**Data availability**

The dataset described in this paper is freely available at Pangaea (Sims et al., 2023) https://doi.pangaea.de/10.1594/PANGAEA.946888. The matchup database of (Land et al., 2023) which is needed to reproduce the algorithm evaluation is available at https://data-cersat.ifremer.fr/data/ocean-carbonate/oceansoda-mmdb/. All the remote sensing datasets needed to create the gridded products are all freely available from their respective online repositories.

**Code availability**

The code used to run this analysis and download the remote sensing datasets is provided in the supplement and is freely available at https://github.com/Richard-Sims/Sims_2023_OceanSODA-UNEXE and Zenodo (Sims and Holding, 2023) https://doi.org/10.5281/zenodo.7863884 . The code can be run on a desktop computer and requires no specialist computing facilities. For example, a laptop with Intel i7-4800MQ 2.70GHz CPU and 8 GB of memory can complete the algorithm evaluation in around one hour. On the same machine the remote sensing datasets take several days to download and reprocess, this is partially subject to local internet speeds and host server speeds. Creating the gridded datasets for each region takes two hours on the same machine.

## Acknowledgements

This work was funded by the European Space Agency via the OceanSODA project (4000112091/14/I-LG). We would like to thank our OceanSODA colleges, Helen Findlay, Luke Gregor and Nicolas Gruber for their helpful discussions and feedback on this work. Hannah Green was supported by a Ph.D. studentship funded by an AXA XL Ocean Risk Scholarship which was awarded to Helen Findlay, Jamie Shutler, and Peter Land.

## Author contributions

Richard Sims led the writing of the manuscript, with inputs from Thomas Holding and Jamie Shutler, and all co-authors made contributions to the final paper. Thomas Holding wrote the majority of the code used for the algorithm evaluation and the creation of the dataset, Richard Sims made additional changes to the code for generating the final dataset. Hannah Green assisted with debugging and testing the code and outputs. Peter Land and Jean Francois provided essential updates to the matchup database. Thomas Holding performed the literature search to identify relevant algorithms from the literature algorithms. Richard Sims produced the final manuscript figures partially based on earlier versions produced by Thomas Holding. Jamie Shutler secured project funding and oversaw completion of the work.

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
