# Peer review of "OceanSODA-UNEXE: A multi-year gridded Amazon and Congo River outflow surface ocean carbonate system dataset"

_Earth System Science Data, 2022_

## Author Comment (AC1)

**Point–by-point responses to editor and reviewers**

Dear Editors and reviewers,

We thank all reviewers for their thoughtful, supportive and constructive comments.

Both reviewers concluded that our manuscript should be published after revision and we have now responded fully to all of the points raised and we will update the manuscript.

The reviewer and editor comments are in black and our responses are in red. In the file containing the revised paper, we will used track changes to show the revised text.

We look forward to hearing from you and thank you for your time.

Best wishes,

Richard Sims and co-authors

**Reviewer 1**

General comments:

A complete and robust dataset on the riverine carbon output and variation is definitely interesting to the ocean carbon community and the international stakeholders. According to the title, the data then tend to present would be highly appreciated by many readers and research communities. However, the manuscript itself have some major flaws, which discourage me to recommend to publish it on its current status. The major aspect concerning me including:

We thank the reviewer for their support and constructive feedback and we are pleased that they highlight that a complete and robust dataset on the riverine carbon output and variation is of interest and use to the ocean carbon community.

We have now addressed all of the reviewer's points and our full replies are below.

1) Language quality is not sufficient and need quite some effort to improve to the level of concise and straightforward academic English.

We will address the sections of the manuscript that the reviewer has identified as needing more concise descriptions. We have also re-read the entire paper and have noted where edits are required to improve readability. In retrospect, we recognise that some of the sentence structures within the manuscript were overly complex. For this reason we will simplify the sentence structures throughout the manuscript.

2) There is a large mismatch between the stated content of the dataset and the actual content of the dataset, specifically, the full carbonate system dataset was said generated, but actually the dataset only consists of the variables DIC and TA.

We are puzzled by this comment. We have double checked and the NetCDF datasets that have been submitted to the data repository contain data for the complete carbonate system (so DIC, TA, pH, aragonite saturation state etc plus combined uncertainties for each parameter).

We assume that the reviewer has overlooked the other content within the NetCDF viewer. To help, the content of the NetCDF files can be easily viewed using the free NASA tool called 'Panoply'.

And the manuscript includes the method used to calculate the complete carbonate system dataset.

3) the description the key method , pyCO2SYS v1.7.1 software, for carbon (pCO2, fCO2) estimate is missing.

Within the methods we describe the pyCO2SYS software version and the settings which were used for these calculations e.g. which dissociation constants (this was given in lines 219 to 224). We did notice that we did not state that the WOA nutrients were used as inputs and so we will correct this. We will also clarify that the input and output temperatures were the same (aka we are not correcting for thermodynamic effects between a lab sample at 25 °C and a typical seawater sample ~10°C). We have referenced the most up to date publication that describes pyCO2SYS, and we will now include a citation for the original methodology as well (Lewis and Wallace 1998).

Lewis, E, Wallace, D, & Allison, L J. Program developed for $CO_2$ system calculations. United States. https://doi.org/10.2172/639712

Specific comments:

1. I did not really understand the processing flow of the data generate, so DIC and TA were estimated with some optimal algorithms and pCO2, fCO2, and pH were the output from the software pyCO2SYS v1.7.1.. but under the title of "full carbonate system data set", TA and DIC are the primary output and took much of the manuscript and evaluation. However, as a reader and a researcher in the ocean carbon community, I would expect pCO2 and fCO2 to be the major variables in the "full carbon system dataset". Please make this one clear.

   The reviewer is correct, DIC and TA were estimated using optimal algorithms and pCO2 and pH were computed using pyCO2SYS. The complete carbonate system can be determined with any two carbonate system parameters (one of which is $pCO_2$ as noted by the reviewer) along with temperature and pressure. But the complex processes controlling pCO2 (e.g. calcification, respiration, photosynthesis) means that estimating $pCO_2$ from commonly observed variables is extremely difficult and challenging. Even with this we would also still

need another carbonate system parameter to calculate the complete carbonate system.

Whereas, in contrast total alkalinity is more tightly linked to salinity, and dissolved inorganic is considered to be linked to salinity and temperature. These properties and the opportunity of exploiting them to observe the carbonate system was first identified by Land et al. (2015) and then evaluated for regions including the Amazon outflow by Land et al., (2019), and this is the reason and justification for the approach that we present.

This explanation is given in the abstract, methods and conclusions on lines 19-21, 22-23, 112-116, 219-222, 510-511 and 512-513. But we will look over these sections again to clarifier this narrative.

2. Throughout the manuscript, the language is not concise or straight forward enough, meaning not academic, and it takes quite some efforts to understand many sentences to grasp their real meaning. And the logic in many of the paragraphs do not really flow.

We appreciate the reviewer's request for the use of straightforward language.

We have now gone through the manuscript and identified sentences with complex structures and we will simplify them without changing the meaning.

3. In the abstract, the author mentioned they generated a dataset of full carbonate system, but the variables they mainly present were TA and DIC. So, there should be a statement on the linkage between the variables in the full carbonate system and the TA & DIC. Or there should be a summary on all variable consist of the dataset and their spatial and temporal resolutions.

Yes, the paper predominantly focuses on the optimal generation of the TA and DIC data and in determining their combined uncertainties. These data, combined with the salinity and temperature data used to generate them, and then allowed us to calculate the full carbonate system parameters (eg. pH, aragonite saturation state). The identified TA and DIC uncertainties (determined using the assessment covered in the paper), along with the published uncertainties for the salinity and temperature data allow the uncertainties in the calculated products (e.g. pH) to be derived (via standard error propagation methods which are included within the carbonate system modelling package, (Humphreys et al., 2022)). All of this information is given within the methods lines 219-226 and lines 234 to 241 of the submitted manuscript.

4. Line26-28, the uncertainty of the TA and DIC were expressed with absolute RMSE and bias. I suggest the percentage uncertainty should be included, i.e., how much the RMSE and bias account for the minima and minimum of the estimated value of TA and DIC.

We understand that giving the uncertainty as a percentage may seem useful but its calculation requires a reference value. However, the large river outflows being studied means that a large range (with steep gradients) in values exists

across the plume, and these values will change with time, so providing the uncertainty as a percentage will be confusing to the reader. We therefore choose to provide minimum, maximum and mean values in the table S4 within the supplementary materials so the user is able to calculate the percentage uncertainty.

We will add a sentence to explain the reasoning for this approach within the paper.

5. line 68-69, "Episodic changes in the carbonate system caused by river plumes can result in financial and biodiversity losses and are of paramount interest to local communities, businesses and policy makers (Doney et al., 2020)." , please give specific examples on what kind of financial and biodiversity loess does it cause and how it is of interest to the stakeholders.

Doney et al., (2020) provide a good overview of the financial and biodiversity losses associated with ocean acidification, these include damage/losses to fisheries, aquaculture, and shoreline protection.

We will include these examples in our manuscript.

6. the first paragraph in section 2.1 is not necessary.

This paragraph defines the key statistics that are used throughout the paper for assessing the uncertainties of the outputs. We feel that it is important to be clear about how the statistics definitions and what they are used for. We also feel that the references to JCGM and GUM are important as this shows that we are following standardized methods for uncertainty analysis.

Therefore we have chosen to keep this text.

7. line 101-102:" This is a clear weakness of comparing wRMSD values from different sources and across differing regions (Land et al., 2019)." If there is a clear weakness of wRMSD, what is the reason to use it to indicate the quality of the dataset?

Apologies for the misunderstanding, this statement was not meant to be a specific criticism of wRMSD as a statistic, but instead we wanted to highlight that should be considered when interpreting the wRMSD values.

We will clarify this point in the revised manuscript.

8. 7line 175, what is Type A uncertainty?

This standardized terminology is defined with the methods section where it says:

"Uncertainty representation and the terminology used throughout this paper are consistent with the International Bureau of Weights and Measures (BIPM) Guide

to the expression of uncertainty in measurement (GUM) methodology (JCGM, 2008)."

By following the GUM methodology we are following the best practice guide for our uncertainty estimate. In GUM a Type A evaluation of uncertainty is defined as "the method of evaluation of uncertainty by the statistical analysis of series of observations".

So a Type A uncertainty assessment is the uncertainty determined by evaluating the dataset using a set of measurements.

9. 219- 226, the brief introduction to the software pyCO2SYS v1.7.1 should be included.

   We will expand the introduction to give a brief introduction to pyCO2SYS.

10, line 244-250, should be in the method section instead of results.

   This paragraph describes the result of the algorithm evaluation and makes references to both figure 1 and table 1 which are part the results.

   There is no new methodological content within this paragraph so we will leave the text as is.

Technical corrections:

1. line 39-41 "The inorganic carbon content of rivers is poorly constrained due to the difficulties of sampling these highly spatial and temporal variable river outflows." The logic is not correct, please revise it.

   Apologies. We will correct this sentence.

   The sentence will now read "The inorganic carbon content of rivers is poorly constrained because it is difficult to sample these large-spatial scale and highly temporally variable river outflows".

2. line 64-65, " River plumes can negatively influence wild fisheries and the aquaculture industry (Mathis et al.,2015;Cattano et al., 2018) as plumes can transport low pH waters that can impact the growth and 65 life stages of many marine organisms (Cai et al., 2021) Additionally," a punctuation is missing.

   The missing full stop after (Cai et al., 2021) will be added.

3. line 102-103," Following the methodology of Land, Findlay et al. (2019) we derive RMSDe from wRMSD,", does not make sense., please revise it.

   The sentence will read as "Following the methodology of Land, Findlay et al. (2019), we calculate RMSDe using the wRMSD result".

4. line 120-125, "To be included in the algorithm evaluation, algorithms needed to be applicable within the…… chlorophyll-a." the sentence is too long to understand, please split it.

Apologies. The sentence will be changed to:

 "The region bounds of the Amazon outflow were defined as being 2° S to 24° N and 70° W to 31° W. The bounds of the Congo outflow were defined as being between 10°S to 4°N and 2° W to 16° E. To be included in the algorithm evaluation, algorithms needed to be applicable to these regions and to take the form of a linear or quadratic relationship with input variables that were easy to obtain and available as spatially and temporally varying datasets. These input variables included sea surface temperature (SST), sea surface salinity (SSS), potential temperature (which is assumed to be approximately equal to SST at the surface), dissolved oxygen (DO), nitrate (NO3-), phosphate (PO4-3), silicate (SiO4-4) and chlorophyll-a."

---

## Author Comment (AC2)

**Point–by-point responses to editor and reviewers**

Dear Editors and reviewers,

We thank all reviewers for their thoughtful, supportive and constructive comments.

Both reviewers concluded that our manuscript should be published after revision and we have now responded fully to all of the points raised and we will update the manuscript.

The reviewer and editor comments are in black and our responses are in red. In the file containing the revised paper, we will used track changes to show the revised text.

We look forward to hearing from you and thank you for your time.

Best wishes,

Richard Sims and co-authors

**Reviewer 2**

The authors reconstructed gridded carbonate system datasets by using a data matchup method based on relationships between carbonate parameters and others which had already been established in the past. While many studies have explored such relationships based on ship-based observations during these decades, the authors utilized these efforts in an effective manner. Such a study is unique and is worth being published, but there are major concerns to be clarified before publication in this journal. I'd like to encourage the authors to improve the study and to revise the manuscript for better understanding.

Thank you for your positive comments and support of our work. We have now addressed all of your comments in full and will revise our manuscript.

**General comments**

Oceanographic characteristics of the studied areas considered, one of the important points of the method is skill to estimate carbonate parameters of low salinity seawaters, which are complexly influenced from both river outflows and heavy precipitation along the ITCZ. On the other hand, relatively higher salinity (S > approx. 34) seawaters in these regions have similar chemical properties to those in the nearest open ocean, where large scale ocean circulations dominate the seawater carbonate chemistry. According to attached supplement files, measurement data used in the matchup process were not necessarily restricted to those of low salinity seawaters. It should be emphasized that the presented method derived more appropriate TA and DIC of low salinity seawaters than others did.

Yes, this is a point that has been covered and evaluated in our previous work (within Land et al., 2019).

We will highlight this important point within our manuscript and explain its impact based predominantly on our findings in Land et al., (2019).

Moreover, secular trends of CO2 were not considered in this study, though time-series reconstructions were addressed. It is needed to show reasonable explanation about that.

Nowadays prevalent machine learning-based methods are used for carbonate system reconstructions; five of the six methods which were cited for evaluating observation-based CO2 sink in the IPCC AR6 assessment used machine learning (Canadell et al, 2021, e.g. Fig. 5.8). It should be explained carefully that this study has some limitation that novel reconstructions cannot be included and legacy of past studies only be used.

We evaluated all the algorithms equitably and as they are presented in the original literature, so we did not modify any relationships as that would be a further substantial amount of work (and was not the focus of this work). We agree that environmental conditions will have changed in the time since these algorithms were derived, e.g. due to oceanic uptake of $CO_2$ and increased freshwater content of the oceans. TA is a conserved quantity in the ocean so is not impacted by the uptake of $CO_2$ but may be lower today in the surface ocean than in the past due to freshwater inputs.

We will now include this point within the introduction of our paper.

We do not use any literature algorithms for $CO_2$ and instead use algorithms to derive DIC; this does mean that the large relative ~ 20 µatm increase per decade in $CO_2$ are much smaller for DIC as a percentage of the total inorganic carbon pool. Theoretically an additional term could be added to each DIC algorithms which accounts for the increase in oceanic DIC since each algorithm was developed. However, such a correction effectively equates to a bias in DIC. Furthermore, to fully account for secular trends, the insitu data would also need to be standardised to a reference year.

We will add a sentence to the paper to explain this point.

As part of the OCEANSODA project (from which this paper has been written), the performance of our approach was assessed against the machine learning OCEANSODA-ETHZ output from Gregor and Gruber (2021) (which is one of the methods that the reviewer refers that was used within the latest IPCC assessment). OCEANSODA-ETHZ is a state of the art machine learning approach which was also recently included in the 2021 global carbon budget estimate (Friedlingstein, Jones et al. 2022). We found that in the Amazon outflow the Gregor and Gruber (2021) TA had a wRMSD of 54.03 µmol kg$^{-1}$ (matchups N=87), whereas the best TA uncertainty in our regionally tuned empirical outputs (given within our manuscript) gives wRMSD 34.97 µmol kg$^{-1}$. This is not surprising as these riverine regionally-specific empirical algorithms were trained on data for these regions whereas the machine learning approaches are trained on global data (which is increasing reduced into sub-regions during training, but it is unlikely to become riverine-outflow region specific). The

machine learning approaches cannot be applied to these riverine regions due to them requiring large datasets for training.

We will add additional text into the discussion to explain why globally applied approaches, including machine learning techniques, are likely to perform poorly on a regional basis.

**Specific comment**

Overall

Unnatural uses of brackets "()" have to be checked.

We will go through the whole paper and we will check all uses of brackets.

P3 71

Before OceanSODA is presented, successive efforts of investigating empirical relationships between TA/pCO2/DIC and other parameters based on observations have to be mentioned here.

We will add a sentence explaining the reviewers point.

P3 L72-76

A brief explanation of OceanSODA is necessary.

We will now define the acronym in the text and briefly introduce the objective of the OceanSODA project.

P4 L103

A brief explanation of RMSDe is necessary.

Agreed. We will include a brief explanation and explain that the full details can be found in Land et al (2019).

P8 L244- Figure 1 Fig. 1 obviously shows that the four selected algorithms have the lowest RMSDe, but doesn't explain whether they are the best even in low salinity regions. It is questionable that Lee et al. 2000; 2006, which propounded global algorithms and (the latter) didn't use salinity as explanatory variables, have the best skill in low salinity Congo basin. This point should be clarified.

Whilst the Lee et. al. (2000;2006) papers provide global algorithms they also provide separate algorithms for different ocean sub-regions. We use these sub-region algorithms and not the global algorithms, and so these sub-region algorithms do use salinity as a predictor.

Within Lee et.al 2000, salinity is not specifically used as a predictor variable however, the relationships are for salinity normalised DIC, so whilst salinity it not a direct input the algorithm, the output is scaled by salinity.

We will add text to explain these points.

Fig. 4, 5, 8, 9

If DICs were successfully reconstructed, trends of increase in DIC and pCO2 and decrease in pH and $\Omega$s would be also derived. The trends are worth being mentioned in the text to support the validity of this datasets.

This is an excellent point and we will include these in the revised manuscript.

Canadell, J. G. et al.: Global Carbon and other Biogeochemical Cycles and Feedbacks. Cambridge University Press, Cambridge, United Kingdom and New York, NY, USA, pp. 673–816, https://doi.org/10.1017/9781009157896.007. 2021

Lee, K. et al.: Global relationships of total inorganic carbon with temperature and nitrate in surface seawater, Global Biogeochemical Cycles, 14, 979-994, https://doi.org/10.1029/1998GB001087, 2000.

Lee, K. et al.: Global relationships of total alkalinity with salinity and temperature in surface waters of the world's oceans, Geophysical Research Letters, 33, L19605, https://doi.org/10.1029/2006GL027207, 2006.